Impact of ocean warming on a coral reef fish learning and memory

Silveira Mayara M. 1
Donelson Jennifer M. 2
McCormick Mark I. 3
Araujo-Silva Heloysa 1
Luchiari Ana C. analuchiari@yahoo.com.br 1
1 Department of Physiology and Behavior, Universidade Federal do Rio Grande do Norte , Natal , RN , Brazil
2 ARC Centre of Excellence for Coral Reef Studies, James Cook University of North Queensland , Townville , Australia
3 School of Science, University of Waikato , Tauranga , New Zealand
Garant Dany
Electronic publication date: 2023 Aug 8
Publication date: 2023
Volume: 11
Electronic Location ID: e15729
Received 2023 Jan 6; Accepted 2023 Jun 18
Copyright: ©2023 Silveira et al.
Copyright year: 2023
Copyright holder: Silveira et al.
License: This is an open access article distributed under the terms of the Creative Commons Attribution License, which permits unrestricted use, distribution, reproduction and adaptation in any medium and for any purpose provided that it is properly attributed. For attribution, the original author(s), title, publication source (PeerJ) and either DOI or URL of the article must be cited.
License URL: https://creativecommons.org/licenses/by/4.0/

Keywords: Temperature, Stress, Behavior, Cognition, Damselfish, Climate change

Funding: Coordenação de Aperfeiçoamento de Pessoal de Nível Superior (CAPES) Conselho Nacional de Desenvolvimento Científico e Tecnológico (CNPq) 306207/2020-6 ARC Future Fellowship FT190100015 This work was supported by Coordenação de Aperfeiçoamento de Pessoal de Nível Superior (CAPES), as a research fellowship to Mayara Silveira and Ana Luchiari. Ana Luchiari was supported by Conselho Nacional de Desenvolvimento Científico e Tecnológico (CNPq) 306207/2020-6. Jennifer M Donelson was supported by an ARC Future Fellowship (FT190100015). The funders had no role in study design, data collection and analysis, decision to publish, or preparation of the manuscript.

==============================
Tropical ectotherms are highly sensitive to environmental warming, especially coral reef fishes, which are negatively impacted by an increase of a few degrees in ocean temperature. However, much of our understanding on the thermal sensitivity of reef fish is focused on a few traits (e.g., metabolism, reproduction) and we currently lack knowledge on warming effects on cognition, which may endanger decision-making and survival. Here, we investigated the effects of warming on learning and memory in a damselfish species, Acanthochromis polyacanthus. Fish were held at 28–28.5 °C (control group), 30–30.5 °C (moderate warming group) or 31.5–32 °C (high warming group) for 2 weeks, and then trained to associate a blue tag (cue) to the presence of a conspecific (reward). Following 20 training trials (5 days), fish were tested for associative learning (on the following day) and memory storage (after a 5-days interval). The control group A. polyacanthus showed learning of the task and memory retention after five days, but increasing water temperature impaired learning and memory. A thorough understanding of the effects of heat stress, cognition, and fitness is urgently required because cognition may be a key factor determining animals’ performance in the predicted scenario of climate changes. Knowing how different species respond to warming can lead to better predictions of future community dynamics, and because it is species specific, it could pinpoint vulnerable/resilience species.

Introduction

Although environmental temperature varies naturally (i.e., seasonally, annually, and diurnally), anthropogenic activities, including CO2 emissions, affect the average range of temperatures experienced globally (Lough, 2012; IPCC, 2018). According to predictions for the next 50–100 years, the Earth’s surface temperature is projected to rise by 2−4 °C (Collins et al., 2013; IPCC, 2018). This warming poses a significant risk to ectothermic species, as their cellular physiology and metabolic rate are governed by the ambient temperature (Brett & Brett, 1971; Schulte, Healy & Fangue, 2011). In addition, tropical stenothermic species (organisms that have a narrow range of temperature tolerance) that have evolved in more thermally stable conditions are likely to suffer from the projected warming due to these species already living close to their thermal maximum (Tewksbury, Huey & Deutsch, 2008; Rummer et al., 2014). Without physiological or behavioral coping mechanisms, many tropical species may reach their thermal limits, affecting population maintenance and the structure and function of ecosystems (Munday et al., 2013).

Fish are the most abundant and diverse group of ectotherms globally. For them, a thermal increase above optimum is perceived as a stressor, which directly stimulates a stress response (Wendelaar Bonga, 1997; Goikoetxea et al., 2021). A fish’s ability to cope with temperature increase depends on the proper response to the stressor, including the activation of the physiological axis, the available energetic substrate to deal with the stressor, and the behavioral changes to confront or avoid the thermal threat (Alfonso, Gesto & Sadoul, 2021). Fish cope with environmental challenges by changing behavior in several ways. These include moving between areas, swimming faster, darting around or staying close together with the shoal to avoid danger (Wolf & Krause, 2014). Threats to these responses brought on by warming water may implicate increased risk to individuals. Studies have shown that several fish species are sensitive to warm temperatures (Jeffries et al., 2016), and pointed out changes in swimming and foraging parameters (Batty & Blaxter, 1992; García-Huidobro et al., 2017; Green & Fisher, 2004; Nowicki, Miller & Munday, 2012), reduction of antipredator response (Lienart et al., 2014), increase in boldness and risk-taking (Angiulli et al., 2020; Davis et al., 2019; Forsatkar et al., 2016), changes in activity level and aggressiveness (Biro, Beckmann & Stamps, 2010; Da Silva-Pinto et al., 2020), and decrease in acoustic communication (Ladich & Maiditsch, 2020). In addition, the rise in temperature causes an increase in the basal metabolism and energy requirements of fish, ultimately impacting their metabolic performance and resulting in changes to their behavior, distribution, and overall fitness (for a review see Brownscombe et al., 2022). To the top of our knowledge, only a few studies showed jeopardizing effects of heating on cognitive functions, such as habitat selection and lateralization in flatfish Solea senegalensis (Sampaio et al., 2016), interest for novelty and navigation ability in Y-maze in zebrafish Danio rerio (Toni et al., 2019), and fear-related response and spatial learning in rainbow trout Oncorhynchus mykiss progeny (Colson et al., 2019). In comparison to our understanding of how the thermal change affects behavior in general (Forsatkar et al., 2016), cognitive responses still need to be adequately investigated.

Considered one of the highest-levels function of the brain, cognition includes the mental capacities that allow animals to understand, learn, solve problems, remember information and adjust behavior (Eichenbaum, 2017; Roy & Bhat, 2016) in order to survive (Moreira & Luchiari, 2022). Learning is the process of gathering new knowledge and changing behavior accordingly; it plays a critical role in adjusting animal behavior through ontogeny so that they are able to properly respond in different situations (Silveira, Oliveira & Luchiari, 2015). In this sense, environmental factors that threaten brain functioning, such as increased environmental temperature, may reduce behavioral and cognitive ability, and ultimately impair the individual fitness (Besson et al., 2017). For aquatic organisms, thermal increase reduces oxygen availability and at the same time increases oxygen demand, reducing neuron activity (Andreassen et al., 2022).

Taking into account the relationship between heat and cognitive aspects, it was demonstrated that higher temperatures reduced information retention in rats exposed to an avoidance task (Batra & Garg, 2005). One explanation is that temperature stress increases the neuroinflammatory response in the hippocampus and impairs memory consolidation (Lee et al., 2015). Recent studies in fish have shown the adverse effects of thermal stress on behavior and cognition, suggesting that increased water temperature reduces expression of synaptic proteins, and impairs cognitive abilities (Angiulli et al., 2020; Nonnis et al., 2021; Toni et al., 2019). Fish have been shown to learn associative tasks in different contexts (for review, see Brown, Laland & Krause, 2011), from a classical visual stimulus and food reward association shown by the ambon damselfish (Pomacentrus amboinensis) (Siebeck, Litherland & Wallis, 2009), to a time-place paradigm based on social group presence performed by zebrafish (De Moura & Luchiari, 2016). Moreover, memory retention was explored in several studies, indicating that fish show the ability to retain memory for hours (De Cognato et al., 2012), days (Silveira et al., 2019), months (Triki & Bshary, 2020; Zion et al., 2011) and in some cases even years (Brown, Laland & Krause, 2011; Tarrant, 1964).

Learning and memory are especially important in spatially complex environments such as coral reefs. Fish can create mental maps with vertical and horizontal surrounding components (Holbrook & De Perera, 2011) and perform associative learning based on visual and non-visual cues (Luchiari, 2016). Any environmental stressor that alters the ability of fish to perform these complex cognitive tasks may impact survival. Most cognitive studies on reef fish have focused the wrasse family due to their role as cleaners and their ability to remember clients (Soares et al., 2017). However, damselfish are key species for reefs as they represent important energetic links between primary and secondary production within tropical coral reefs. In a previous study, we observed that the damselfish Stegastes fuscus could remember where it had an agonistic encounter based on visual color cues (Silveira, Oliveira & Luchiari, 2015), and they recognize and remember a conspecific neighbour for more than 10 days (Silveira et al., 2021). This same species learn to associate the side of the tank with a positive (conspecific) and a negative (shock) reinforcement and remember it for up to 15 days (Silveira et al., 2019).

The predicted impact of ocean warming for many organisms on coral reefs that live close to their thermal limits are dire (Timmers et al., 2021), but little is known on the impact of thermal stress on the cognitive processes that underlie behavioural decisions and survival. The current study investigated the effects of warming on spatial learning and memory formation in a common species of reef damselfish, Acanthochromis polyacanthus. Based on global warming predictions and the tolerance of our focal species, we subjected it to the summer average and elevated water temperatures (control: 28–28.5 °C, moderate: 30–30.5 °C and high: 31.5–32 °C), trained it in a spatial task, and evaluated memory after a 5-day interval. Based on previous research, we expected that A. polyacanthus would have advanced spatial cognitive skills. With elevated water temperature we expected a reduction in learning (measured by behavioral changes over training days, correct choices and time with reward) and a detrimental impact on memory (information retrieve after a time interval). The current variation in environmental temperature impacts all levels of life in the planet, and understanding the thermal variation effects on cognition may generate results to be applied from veterinarian to ecological and management fields.

Material & Methods

Ethics statement

All fish maintenance and experimental protocols were approved by The Animal Ethics Committee of James Cook University, Townsville, Queensland, Australia (A2693).

Study species

The spiny chromis damselfish, Acanthochromis polyacanthus (Bleeker 1855), is common on inshore and offshore coral reefs in the Indo-Australian archipelago (Lieske & Myers, 1994). Adult A. polyacanthus form monogamous pairs and breed primarily during the summer months (Robertson, 1973). Joint parental care is provided to benthic eggs and direct developing juveniles for up to 30–45 days post-hatching (Kavanagh, 2000; Pankhurst, Hilder & Pankhurst, 1999). As mature adults A. polyacanthus are highly territorial (Pankhurst, Hilder & Pankhurst, 1999) of their home range that may be up to 24.8 m2, but 6.8 m2 on average (Cowlishaw, 2014).

Acanthochromis polyacanthus present a critical thermal minimum at 15.5 °C and maximum at 38 °C, with maximum growth and weight gain at 28 °C (Zarco-Perelló, Pratchett & Liao, 2012). Temperatures above 34 °C led the fish to die within 8-15days (Zarco-Perelló, Pratchett & Liao, 2012). Being a reef-associated and non-migratory species that guard and aerate eggs, the spiny chromis damselfish is highly susceptible to any environmental change affecting reefs.

Animal collection and husbandry

Adults of A. polyacanthus were collected from reefs around Cairns (northern region of the Great Barrier Reef) Queensland, Australia by professional collectors (Cairns Marine) to compose the stock population (30 animals). These animals were maintained on an outdoor recirculating system (∼30,000 L) at James Cook University’s Marine and Aquaculture Research Facility (MARFU). Pairs of fish (one male and one female) were kept in 60 L white plastic round tanks, with half terracotta pots to serve as both shelter and nesting sites. Pairs were fed twice a day ad libitum with aquaculture manufactured INVE NRD G20 pellets (Protein minimum 55% and fat minimum 9%). Water temperature was kept stable at 28-28.5 °C and salinity ∼35 ppm.

Before experimental phase begin, 30 fish were treated as per disease prevention protocol because fish were changing systems (250 ppm formalin, 37% w/w formaldehyde for no less than 45 min). Then, these fish were transferred to an experimental temperature-controlled room in individual 32 L tanks (432 × 324 × 305 mm) with a PVC pipe on the bottom of each aquarium for shelter to reduce the captivity stress. All the tanks had constant water flow from three partially exchanging systems (3 × 8,000 L), each with its own temperature control (Heat pump, Toyesi Titan 15 kw), UV sterilization and mechanical filtration. Throughout the experiment salinity was maintained at ∼35 ppm and all tanks were provided with constant aeration. To minimize additional interactions with fish, other than the behavioral testing, tanks were cleaned only when fish were not present in the tank (i.e., in the behavioral tests). Fish were fed 1–2 times daily with aquaculture manufactured INVE NRD G20 pellets. The experimental aquarium room was maintained in 12:12 photoperiod (light: dark). Fish were kept in the described condition and water temperature was maintained at 28–28.5 °C for 6 days (i.e., acclimation) before warming conditions started.

Warming exposure

After acclimation period, fish were divided by chance into one of three temperature groups: a control group maintained at 28–28.5 °C (n = 10), a moderate warming group maintained at 30–30.5 °C (n = 10), and a high warming group maintained at 31.5–32 °C (n = 10). A. polyacanthus has been well studied in relation to physiological thermal sensitivity. Multiple studies have shown a critical thermal maximum (the temperature at which cellular components and process become disrupted) from 36.0–37.5 °C in populations that naturally experience summer water temperatures of 28–30 °C (Rodgers et al., 2018; Zarco-Perelló, Pratchett & Liao, 2012; Clark et al., 2017). However, impacts to performance traits including aerobic metabolism (Nilsson et al., 2009; Donelson & Munday, 2012; Rodgers et al., 2018), growth (Munday et al., 2008a; Munday et al., 2008b; Zarco-Perelló, Pratchett & Liao, 2012), reproduction (Donelson et al., 2010), and mortality (Rodgers et al., 2018) can occur at much lower temperatures only 1–3 °C above summer from 30–32.5 °C.

For the two elevated treatments used here, the water temperature was increased at a rate of 0.5 °C per day (i.e., moderate warming = 4 days, and high warming = 7 days). After reaching the desired temperature, fish remained in the warm conditions for two weeks before the beginning of the experimental protocol.

Experimental protocol

The apparatus used in this phase was a square tank with four grey PVC walls (90 × 90 cm) forming four halls (Fig. 1). A small transparent tank was placed at the end of each hall (20 × 20 cm). One of these tanks received a stimulus fish while the other three were empty. On the hall entrance where the stimulus fish was, a blue tag was displayed as a cue. The protocol used and the choice of a conspecific as a stimulus were based on previous studies using the dusky damselfish Stegastes fuscus (Silveira et al., 2019; Silveira, Oliveira & Luchiari, 2015). A start box (opaque PVC tube of 15 cm diameter) was added to the center of the tank, and experimental fish was placed there before each training and test session.

Figure 1 Schematic views of the tank used for the learning tests in Acanthochromis polyacanthus and Amphiprion percula.

The numbers show the dimensions of the tank in cm. All walls were gray and one of them received a blue card to indicate the hall where the conspecific fish was placed. The start box (10 cm diameter) was placed in the center to serve as a release area. (A) Overview, (B) side view.

All experimental phases were carried out using water from the system where fish were housed (same temperature). For that, the experimental tank was placed inside a bigger tank connected to the recirculating system that kept water quality and temperature constant (i.e., acting as a water bath).

The experimental procedure was divided into four phases: (1) habituation to the apparatus, (2) associative learning training, (3) learning test, and (4) memory test. After each individual trial in the experimental apparatus, fish were returned to their housing tank until their next trial.

1. Habituation (days 1–4): Animals were placed individually in the start box for 30 s, after which they were released to explore the apparatus. There was no colored cue card and no stimulus fish in the tank. On days 1 and 2, each fish explored the tank for 15 min. On day three fish were allowed to explore the tank for 10 min. On the last habituation day, fish were given 5 min to explore the tank. This procedure was conducted to reduce novelty stress.

2. Training—Associative learning (days 5–9): In this phase one of the tank walls displayed a blue tag as a cue to indicate the entrance of the hall where a stimulus fish was located. Animals were individually placed in the start box for 30s and then released for exploration of the tank for 5 min. The procedure was repeated four times each day with a 90s interval between runs (total of 20 training runs). The hall where the stimulus fish was placed was maintained the same in all training trials.

3. Learning test (day 10): There was no stimulus fish in the tank in this phase, but the blue cue tag was present. After the last associative learning trials, fish were tested to determine whether they had learnt the position of the stimulus fish. For this, each animal was positioned in the start box for 30s and then released to explore the tank for 10 min, similar to what had occurred in the training trials; however, there was no stimulus fish.

4. Memory test (day 15): In this phase, no stimulus fish were present but the blue cue tag was present. After the learning test, fish were left for a 5-days interval in its housing tank before the memory test was performed. The same protocol was followed: fish stayed 30s in the start box and then explored the tank for 10 min.

All fish went through the four phases above described. The order each fish from each experimental group (control, moderate warming and high warming) entered the experimental tank was randomized every day. No other strategy to control confounders were used. All experimental steps were recorded on video and analyzed using the ZebTrack software, to extract the following variables: time spent in the areas; latency for arrival in the target hall (i.e., blue zone with stimulus fish); correctness of the first choice (i.e., target hall) average speed; distance covered; and freezing. Learning was inferred by the reduction in time to enter the target hall over the training trials and during the learning test, the increase in time spent with the reward during the training trials and in the target hall during the learning test, the number of training trials needed to find the reward, and the correctness of the first choice (target hall) after fish was released in the apparatus. Memory was accessed after 5-days interval by the latency to enter the target hall, correctness of the first choice and time spent in the target hall on the memory test.

As the tracking software is automatized, no blinding was needed for the analysis. Fish used as the stimulus were randomized every day with respect to the experimental fish, so they did not see the same fish during the experimental days. The stimulus-fish were conspecific to the focal fish and were not trained in the maze. This study did not need humane endpoints.

Statistical analyses

All data were analyzed for homogeneity, normality, and possible outliers, as suggested by Zuur, Ieno & Elphick (2010). Exploratory analysis indicated no need to exclude animals from the dataset. The Akaike Information Criterion (AIC) was used to choose the most appropriate model indicating greater quality and simplicity with lower AIC values. After that, mixed models and post hoc Bonferroni tests were performed. Then, data regarding the time fish spent in each maze area were transformed using squared root to adjust for gamma distribution. This distribution was also used for latency to enter the target hall and locomotor parameters. These variables were considered quantitative.

For the time that each fish spent in different areas of the tank, we considered as a fixed effect the variables learning test (two levels: learning and memory) and maze areas (two levels: Halls and Target). Individuals were considered as random effect. For latency to enter the target during associative learning, treatments (three levels: control, moderate warming and high warming) and training days (five levels: days 1, 2, 3, 4 and 5) were considered as fixed effect, and individuals as random effect. For latency during the learning and memory test and also for the locomotor parameters, treatments (three levels: control, moderate warming and high warming) and test days (learning test and memory test) were considered fixed effects as individuals were the random effect.

Concerning hall choice, the normal distribution was used in which the percentage of entry in the target hall was considered the quantitative dependent variable. Treatments (control, moderate and high warming), associative learning days, learning test and memory test were considered fixed effects. Individuals were considered a random effect.

To build all the mixed models we used the glmer or lmer command (lme4 package). Data were analyzed using ‘tidyverse’, ‘ggpubr’, ‘lme4’, ‘bbmle’, ‘car’, ‘multcomp’, ‘emmeans’, and ‘ggally’ packages (Wickham & Wickham, 2017; Kassambara & Kassambara, 2020; Bates et al., 2009; Bolker, 2017; Fox et al., 2012; Hothorn et al., 2016; Lenth et al., 2019; Schloerke, Crowley & Cook, 2018) from R (RStudio version 4.2.2 for Windows; RStudio Team, 2022). For all comparisons, the significance level was set at p < 0.05. R scripts are available as Supplementary Material.

Results

The percentage each fish spent in different areas of the tank during the learning test and the memory test in A. polyacanthus was explored for each temperature group (Fig. 2). Comparisons made by the mixed model indicated statistical significance between the preference for the target zone in contrast to the percentage of time in the hall area. This difference can be seen for the control (Fig. 2A; GLMM, χ2=9.91, df = 1, p = 0.001) and moderate warming (Fig. 2B; GLMM, χ2 = 4.57, df = 1, p = 0.032). For the high warming treatment, the individuals did not distinguish between the hall and target areas (Fig. 2C; GLMM, χ2 = 1.92, df = 1, p = 0.16). The post hoc test of Bonferroni depicted that the control group spent more time in the target hall on both learning and memory tests than the moderate and high warming group (p = 0.001), while high vs. moderate warming groups differed only on the learning test when the moderate group spent more time on the target hall (p = 0.032; Fig. 2).

Figure 2 Time spent in the non-target and the target halls of the tank during learning and memory test.

Acanthochromis polyacanthus time spent in the non-target halls (mean of hall 1, 2 and 3) and the target hall of the tank during the learning test and the memory test. The data show the percentage of time spent (n = 10). The central area was not considered for the analysis. The target hall was presented with a blue card (stimulus) to orientate the fish. During the training trials, the hall with the blue card showed a conspecific fish to serve as social reward. Fish were allowed to explore the tank for 10 min per day. The learning test (gray bars) was applied after 20 training trials and no reward was presented. The memory test (blue bars) was applied 5 days after the learning test. A. polyacanthus were held at three different water temperature (A) 28–28.5 °C –control, (B) 30–30.5 °C, and (C) 31.5–32 °C. Different lower-case letters indicate statistical significance (a ⁄ = b) at p < 0.05 (GLMM).

The latency to enter the target hall is presented in Fig. 3. Note that panel a shows horizontal box plots with data points from each training day. Figure 3A allows us to compare the characteristic of several individual observations referring to latency and days of associative learning (fixed effects) in a set of numerical variables (with different units) grouped to the same scale. Values are plotted as a series of connected lines on each axis, allowing every data element of the dataset in use to be represented. The mixed model showed no statistical significance for temperature over the days of associative learning (Figure 3A; GLMM, χ2 = 2.87, df = 4, p = 0.57) or between treatments (Fig. 3A; GLMM, χ2 = 3.86, df = 2, p = 0.14). On the other hand, during the learning and memory tests (Fig. 3B), temperature changes affected individuals in the moderate and high warming treatments compared to the control (Fig. 3B; GLMM, χ2 = 121974.4, df = 2, p < 0.0001) and between learning and memory test (Fig. 3B; GLMM, χ2 = 1041.8, df = 1, p < 0.0001). The post hoc Bonferroni comparison test showed the three temperature treatments differed from each other (p < 0.0001).

Figure 3 Latency to enter the target hall over the training days.

Acanthochromis polyacanthus latency to enter the target hall during the training days, and the learning and the memory test after being held at three different water temperature control: 28–28.5 °C, moderate warming: 30–30.5 °C, and high warming: 31.5–32 °C. Data are shown as median and quartiles (n = 10). (A) During the training days (four trails/fish/day), the target hall was presented with a blue card to orientate the fish and a conspecific served as social reward in the target hall. An asterisk (*) indicates the median value of the box plot. Latency and days were scaled to values between 0-1. (B) During the learning and memory tests, no reward was present but the blue card guided fish behavior. Fish latency to enter the target hall is presented for the learning test and memory test with a line connecting the same individual. Different lower-case letters above plots indicate statistical significance (GLMM) at p < 0.05 (a ⁄ = b ⁄ = c).

Regarding the percentage of first choice for the target hall, the linear mixed models showed statistical significance between treatments (Fig. 4; LMM, χ2= 33.05, df = 2, p < 0.0001). Bonferroni’s post hoc comparison test showed the control differed from moderate warming and high warming groups (control vs. moderate: p = 0.003; control vs. high: p = 0.0004; moderate vs. high: p = 0.40). LMM indicated no statistical significance over the days (Fig. 4; LMM, χ2 = 6.18, df = 6, p = 0.40).

Figure 4 Percentage of first choice for the target hall in the experimental tank.

Percentage of Acanthochromis polyacanthus’s first choice for the target hall of the tank during the training days, learning test and the memory test. Fish were held at three different water temperature: (A) 28–28.5 °C (control), (B) 30–30.5 °C (moderate warming), and (C) 31.5–32 °C (high warming). The target hall was presented with a blue card (stimulus) to orientate the fish. During the training trials (four trails/day), the target hall presented a blue card and a conspecific fish inside the target hall served as social reward. Fish were allowed to explore the tank for 10 min per day. The learning test was applied after 20 training trials and no reward was presented. The memory test was applied 5 days after the learning test. Data are shown as percentage of correct first choice (target hall) over the experimental phases. During the training days, each fish was tested four times. During the learning test and memory test, each fish was tested only once (n = 10). Different lower-case letters indicate statistical significance (LMM) at p < 0.05 (a ⁄ = b).

The locomotor parameters obtained from the damselfish A. polyacanthus exposed to different water temperatures are shown in Fig. 5. Comparisons made by the mixed model showed that average speed was not significantly changed by temperature (GLMM, χ2 = 0.18, df = 2, p = 0.91; Fig. 5A), test day (GLMM, χ2 = 2.00, df = 1, p = 0.15; Fig. 5A) or the interaction terms temperature vs. test days (GLMM, χ2 = 3.10, df = 2, p = 0.21; Fig. 5A). For maximum speed, mixed model indicated no statistical significance between water temperature groups (GLMM, χ2 = 0.15, df = 2, p = 0.92; Fig. 5B), test day (GLMM, χ2 = 0.74, df = 1, p = 0.39; Fig. 5B) and terms of interaction (GLMM, χ2 = 1.65, df = 2, p = 0.44; Fig. 5B). The mixed model test for total distance traveled showed no statistical significance for temperature (GLMM, χ2 = 0.16, df = 2, p = 0.92; Fig. 5C), test days (GLMM, χ2 = 1.98, df = 1, p = 0.16, Fig. 5C) and interaction between temperature and days (GLMM, χ2 = 3.18, df = 2, p = 0.20; Fig. 5C). For time stopped, the mixed comparison model showed that there was no statistical significance for temperature (GLMM, χ2 = 0.17, df = 2, p = 0.92), but it was statistically significant for test day (GLMM, χ2 = 12.12, df = 1, p = 0.0005; Fig. 5D) and the interaction temperature vs. days (GLMM, χ2 = 8.49, df = 2, p = 0.014; Fig. 5D). Bonferroni’s post hoc test showed that control group presented less time stopped during the memory than the learning test, while it did not differ from the moderate and high temperature groups (Fig. 5D).

Figure 5 (A–D) Locomotor parameters during the learning and memory tests.

Effects of temperature (28–28.5 °C –control, 30–30.5 °C –moderate warming, and 31.5–32 °C –high warming) on Acanthochromis polyacanthus locomotor parameters during the learning test and the memory test. Data correspond to median values and quartiles (n = 10). Fish behavior was recorded for 10 min. No statistical significance was observed between groups (GLMM, p > 0.05).

The raw data of this study are available as Supplementary Material.

Discussion

We observed that water temperature affects reef fish learning of a spatial task and the retention of this information as memory. The coral reef fish Acanthochromis polyacanthus exhibited learning and memory ability of an associative task with social reward (conspecific fish), with fish held at control temperature (28–28.5 °C) able to learn the task and remember 5 days later. However, memory of the task was compromised at moderate warming (30–30.5 °C), and no learning occurred at high warming (31.5–32 °C). Moreover, A. polyacanthus showed conserved locomotor parameters under warmer conditions. It is the first time reef fish have been tested for the effects of rising temperature on cognition, and our results suggest that warming impairs A. polyacanthus performance, both in terms of learning and memory.

Under normal summer conditions, A. polyacanthus exhibit the ability for associative learning and memory, as has been seen in other damselfish species (Silveira, Oliveira & Luchiari, 2015; Silveira et al., 2019). In the present study, A. polyacanthus explored the tank and could associate a cue (a blue card at the hall entrance) with the presence of a conspecific over 20 training trials, which is observed by the reduced latency to enter the target hall over trials (Fig. 3) and the higher percentage of first choice for the target hall during the trials (Fig. 4). This spatial memory is perhaps unsurprising as adult A. polyacanthus are seen in pairs exploring and defending an average of 6.8 m2 sized territorial area (Cowlishaw, 2014). Moreover, A. polyacanthus showed memory retention of the task after 5 days, when it reduced latency to enter the hall where the conspecific was expected to be found. Learning and memory are mental processes that allow animals to obtain information and respond accordingly. Learning to navigate and remembering where to go and when affects an individual’s performance, especially in spatially complex reef environments inhabited by many different species (Graham & Nash, 2013). These cognitive abilities favour animals flexibly adjusting behavior and adaptively changing their phenology to cope with stochastic changes in their environment.

However, we found that ocean warming was detrimental to learning and remembering an associative spatial task. With moderate warming of 30–30.5 °C, A. polyacanthus were still able to learn but memory was compromised, and the fish could not accomplish the task 5 days later. Fish latency to enter the target hall (Fig. 3B) is similar to those of control fish. However, with high warming of 31.5–32 °C, no learning could be detected after the same number of training trials (increased incorrect first choices and latency to enter the target hall).

Our results indicated that fish took longer to enter the target zone (Fig. 3), made less correct choices (Fig. 4), and spent less time in the target hall (Fig. 2) when A. polyacanthus were maintained in moderate or high temperature for two weeks. Different from the control fish, the warm-acclimated fish took longer to first discover the target hall during the learning phase. This implies that these fish would need a longer training period to reach the same level of choice correctness observed for the control fish. If we consider that locomotion was not the factor affecting these differences (as shown in Fig. 5), what may have driven these differences relies on the animal’s ability to acquire and integrate new information. The more trials or repetitions it takes to learn something, the more cognitive effort and processing are required (Oliveira, 2013). Fish that show a propensity to learn and adapt to new situations are more likely to survive and thrive in their environment (Salvanes & Braithwaite, 2006). The fish’s capacity to adjust behavior based on previous experiences and outcomes indicates its cognitive flexibility, while cognitive efficiency involves the time required to process information and extract relevant patterns or concepts (Shettleworth, 2001; Dukas, 2004; Ebbesson & Braithwaite, 2012). Most fishes adjust their behavior according to environmental demands, what help then to cope more effectively (Ebbesson & Braithwaite, 2012). However, it is known that prolonged exposure to stress reduces brain processing, and may disrupt attention and exploration, which are fundamental factors for learning (Warburton & Hughes, 2011). In this sense, warming may not impair learning /memory directly but may affect processes that facilitate it, demanding more time and effort from the animals to achieve the same results.

Prolonged stress, as global warming imposes, can exceed the regulatory capacities of the organisms, thus demanding more energy for allostasis and decreasing energy for other functions including cognition (Alfonso, Gesto & Sadoul, 2021; Bradshaw, 2003; Wendelaar Bonga, 1997). At the same time, the cognitive functions related to perception, decision making, learning, and memory are required to deal with stressful situations (Shettleworth, 2009), and the impairment of these functions can be highly deleterious (Danner, Coomes & Derryberry, 2021; Soravia, Ashton & Ridley, 2021). Moreover, the stress due to warming reduce oxygen availability to the brain (Andreassen et al., 2022) and increase neuroinflammatory response in the hippocampus, which can lead to memory loss, impaired neurogenesis, and neuronal death (Lee et al., 2015).

Alterations caused by temperature increase on behavioral plasticity are essential factors to be investigated since adjusting and formulating new responses to adverse situations are central to an individual’s performance (Soravia, Ashton & Ridley, 2021). It is striking that we observed such strong effects on cognition with only 14–24 days exposure to a 2–4 °C increase, as similar condition have already been experience on the Great Barrier Reef with marine heatwave events (Hughes et al., 2018a; Hughes et al., 2018b). It is probable that the warming that will occur in the near future will affect cognitive responses involved in learning and memory. The extent of this impact is likely to be greater with a higher degree of temperature increase.

This finding of thermal impairment is in agreement with previous work on a number of other fish species. A pioneer study by Cerf & Otis (1957) showed adverse effects of 7 °C heating on long-term memory of temperate goldfish (Carassius auratus). Further research on temperate and cold water fish species has shown similar detrimental effects of elevated temperature on cognition (Colson et al., 2019; Sampaio et al., 2016; Toni et al., 2019) as has been observed for other traits (e.g., aggression; Da Silva-Pinto et al., 2020). Here, we show extreme thermal sensitivity in our tropical fishes, with a small thermal increase of only 2 °C and for a shorter duration of 14 days compromising learning and memory performance.

Our study selected temperature increases that could occur in the near future with marine heatwaves (Hughes et al., 2018a; Hughes et al., 2018b) and that could become common summer temperatures in the future with global warming, as a starting point for determining species’ tolerance. However, our study species, A. polyacanthus, is well known to exhibit greater thermal plasticity with development from early life or across generations at elevated temperatures, allowing restoration of function back to control levels (Donelson & Munday, 2012; Donelson et al., 2011; Donelson & Munday, 2015). Future research should investigate whether the negative effects to cognition can be restored with early development and across generational exposure.

Due to global warming, many studies have investigated the physiological effects of heating in fish performance, such as on growth (Munday et al., 2008a; Munday et al., 2008b), reproduction (Hilder & Pankhurst, 2003), heat shock proteins (Schulte, Healy & Fangue, 2011), aerobic scope (Killen et al., 2014), escape reaction (Warren, Donelson & McCormick, 2017) and so on. However, the impact of warming on cognition is still a topic poorly explored. Understanding the relationships between heat stress, cognition and fitness is urgent because cognition may be a critical factor in understanding how climate changes affect animal performance (Soravia, Ashton & Ridley, 2021). Heat stress can lead to physiological changes such as increased oxygen demand, increased heart rate, and reduced blood flow to the brain, which can impair cognitive abilities as attention, memory and decision-making. Our study demonstrated that temperature increase has negative impacts on cognitive skills for one species, however, the prevalence more broadly remains unknown. Therefore, understanding the effects of heat stress on cognition and fitness is important for maintaining optimal performance and suggesting conservation actions. Our findings underscore the need for further research investigating the effects of increasing temperature in different species, using other thermal exposures (warming, timing and length) and learning tasks, and exploring the interactive effects of temperature with other environmental stressors, such as acidification and pollution.

Conclusion

In this study, we hypothesized that the damselfish A. polyacanthus would learn and remember the spatial task, but that water temperature would reduce the fish’s performance. Indeed, A. polyacanthus learned the task during the 20 training trials and showed reduced latency to enter and higher occupation of the target area of the maze both on the learning test and the memory test executed 5 days later. However, increasing the water temperature to 1.5–2.0 °C still allowed learning but compromised the fish memory after a 5-day interval, while rising water to 3.0–3.5 °C jeopardized both learning and memory of the damselfish. The negative impacts of warming on fish cognition highlight the need for understanding how wide these effects can be, that besides interfering with animals’ physiology, may implicate several harmful consequences for the community dynamics. More than that, ways to mitigate rising water temperature are urgently required.

Supplemental Information

Supplemental Information 1 ARRIVE Checklist.

Click here for additional data file.

Supplemental Information 2 Raw data.

Data collected during the experimental phase with A. polyacanthus held at three different water temperature: 28–28.5 °C (control), 30–30.5 °C (moderate warming), and 31.5–32 °C. (high warming).

Click here for additional data file.

Supplemental Information 3 R script for Fig. 2

R script used to compare the time spent in non-target and target hall (results presented in Fig. 2).

Click here for additional data file.

Supplemental Information 4 R script for Fig. 3.

R script used to compare the latency to enter the target hall throughout the experimental phase (results presented in Fig. 3).

Click here for additional data file.

Supplemental Information 5 R script for Fig. 4.

R script used to compare the first choice of the fish for a hall over the exerimental phase (results presented in Fig. 4).

Click here for additional data file.

Supplemental Information 6 R script for Fig. 5.

R script used to compare the locomotor parameters of the fish (results presented in Fig. 5).

Click here for additional data file.

We would like to thank the staff team of Marine and Aquaculture Research Facility (MARFU) and the ARC Centre of Excellence for Coral Reef Studies, James Cook University for their support of the experiments. Thanks to Blake Spady for construction of the behavioral testing tank, and to Francisco Diego and Vanessa Augusta for technical assistance with video analysis.

Additional Information and Declarations

Competing Interests

Author Contributions

Animal Ethics

Data Deposition

The authors declare that there are no competing interests.

Mayara M. Silveira conceived and designed the experiments, performed the experiments, authored or reviewed drafts of the article, and approved the final draft.

Jennifer M. Donelson conceived and designed the experiments, authored or reviewed drafts of the article, and approved the final draft.

Mark I. McCormick conceived and designed the experiments, authored or reviewed drafts of the article, and approved the final draft.

Heloysa Araujo-Silva analyzed the data, prepared figures and/or tables, and approved the final draft.

Ana C. Luchiari conceived and designed the experiments, authored or reviewed drafts of the article, and approved the final draft.

The following information was supplied relating to ethical approvals (i.e., approving body and any reference numbers):

The Animal Ethics Committee of James Cook University, Townsville, Queensland, Australia approved this research (A2693).

The following information was supplied regarding data availability:

The raw data show all fish behavioral parameters and are available Supplemental Files.

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
