# Peer review of "Impact of ocean warming on a coral reef fish learning and memory"

_PeerJ, doi:10.7717/peerj.15729_

## Round 0.1 · original submission · Major Revisions

We have received three reviews for your manuscript. All reviewers found merits in your study but also raised a number of important points that deserve further revisions.

In particular, all reviewers emphasized that statistical analyses were unclear and needed to be detailed further - providing R scripts was also suggested.

Other important points raised by reviewers concerned the criteria used to define learning as well as the training protocol used, which were deemed questionable. Other factors may be at play here (instead of learning) and may have hampered the conclusions of the study.

Finally, reviewers also recommended that figures (and presentation of results in general) should be improved.

Reviewer 1 ·

Excellent Review

This review has been rated excellent by staff (in the top 15% of reviews)
EDITOR COMMENT
Very helpful and constructive review.

Basic reporting

Basic Reporting – This manuscript was clear and well written, provided relevant context and citations, and conforms to the standards of the journal.

Experimental design

Experimental design – This work is within the scope of the journal, and the research question is well-defined, relevant, and fills a clear knowledge gap. However, I have some concerns about the experimental approach (see detailed comments below).

Validity of the findings

Validity of the findings - All data are provided, but the analyses are not described in detail. The links between results and conclusions are not always clear, and I think more needs to be done to interpret the data with more nuance (see detailed comments below).

Additional comments

This manuscript was well written and enjoyable to read. The question of how fish cognitive performance will fare in a warming world is highly relevant and fills a clear knowledge gap. The experimental approach was quite standard for this type of investigation, and the data appear to generally be of high quality. However, I have some concerns about the data analysis and interpretation of the results that may affect the main conclusion of the paper that warm conditions impair cognition.

This paper fundamentally revolves around the idea of “learning”, but the term is never clearly defined by the authors and it is not totally clear what results would constitute evidence for/against learning. At the end of the introduction (line 108) the authors suggest that a reduction in learning rate is a key indicator of learning impairment, but this is not the only criterion of learning used in the paper and the term “learning rate” does not appear anywhere else in the manuscript. One way to test for differences in learning rate would be to evaluate the effect of “day” in the mixed models used by the authors. The only response variable influenced by “day” was latency to enter the target hall (e.g. line 225), and this appeared as an interaction between day and temperature treatment. In fig 3, it appears that this interaction term reflects the shortening latency over time in the control group, but no change in latency in the warming groups. This to me is by far the best evidence for the main conclusion of the paper that high temperatures may impair learning. Control fish learn but fish exposed to warming fish don’t appear to learn, if learning is defined as change in latency over time. Results from the other main response variable measured, time spent in each hall, do not show nearly the same evidence for learning. The authors suggest that because control fish spent more time in the “target” hall than the warm-exposed fish, this is evidence of associative learning. However, the effect of “day” was highly non-significant (p=0.91, interaction p=0.41) which strongly suggest to me that no learning took place (i.e. performance didn’t improve over time). Furthermore, control fish spent ~28% and 26% of their time in the “target” hall, just slightly over the 25% expected by chance given that there were four possible halls to choose. Thus, I do not understand how these results show evidence of learning in any of the treatment groups, let alone learning impairment at high temperatures. I urge the authors to think hard about which data support their conclusions, and about how clear-cut of a story they present.

I am also slightly worried about the training protocol used by the authors. Fish were trained by allowing them to swim around the tank 20 times for 5 minutes each time, which was supposed to allow them to associate a blue tag with the presence of a conspecific. However, there is no information given about how long it took fish to initially find the “target” hall, and looking at the raw data it appears that 5 fish never found the target hall in any of the 20 trials. How can learning be assessed if these fish never had the opportunity to associate the blue tag with a conspecific? For other fish, it is not reported how many trials it took before the conspecific was located. In many of these cases, the test animals may well have had fewer than 20 training trials in which an objective could be “known” to the test fish. In these sorts of learning experiments, it is common practice to initially allow the fish to explore until the cue/reward combination is discovered so that subsequent trials can more robustly assess the strength of the association learned between these two. While this is not the case here, perhaps the authors could include the number of trials it took for the test fish to find the “target” as a covariate in their models to account for this variation in learning opportunity (for example, I see that control fish 3 took 4 trials to find the target hall). I do not think it is reasonable to include the 5 fish that never found the target hall in the analysis. Finally, I think the dots in Figure 3 need to be jittered to a larger width to indicate the number of 300s latency datapoints in the moderate and high warming groups – these are currently obscured and I was initially quite confused by the apparent low number of datapoints.

Minor comments
Line 54 – It would be helpful to further explain the physiological effects of warming on the fish brain (and thus presumably cognition) in the text here. For example, it is generally thought that increased baseline metabolism that results from warming affects the energy budget of fishes in complicated ways (see Borwnscombe et al 2022 J Fish Biol https://doi.org/10.1111/jfb.15066 for a nice recent review), and recently it has been suggested that the brain dysfunction at high temperatures may ultimately determine the overall thermal tolerance of fishes (Andreassen et al 2022 PNAS https://doi.org/10.1073/pnas.2207052119).

Lines 94-98 – This is an important point, but it seems out of place here. Consider moving this paragraph to just before the paragraph about damselfish as model species that begins on line 85.

Line 166 – the methods describes two “habituation” periods, one that occurred before warming occurred (lines 143 and 146), and one during the training experiment (line 166). I suggest re-naming one of these for clarity (e.g. call the first period acclimation).

Line 205 – The statistical methods are unclear from this brief description. Please add further detail, or even better would be to include the R script as supplemental material.

Line 261 – what evidence supports this statement. Please see comment above, I think this is only supported by the latency data.

Line 274 – again, please indicate the evidence used to support this conclusion.

Line 297 – In the introduction it is suggested that there is little known about the effects of temperature on fish cognition, but several key references are included here. I suggest referring to these studies in the introduction – they do not negate the value of the current work, but are helpful for providing context.

Finally, I would encourage the authors to provide raw data on all plots (as they have in Figure 3), or even better to use plots that show the distribution of their data with more nuance than bar graphs. A helpful primer is provided by Weissgerber et al 2015 (https://doi.org/10.1371/journal.pbio.1002128)

Reviewer 2 ·

Basic reporting

Clear and unambiguous, professional English used throughout.

The manuscript is well written and easy to read. The English level is high.

Literature references, sufficient field background/context provided.

The introduction is clear and detailed. I suggest authors to add more recent publications in the literature, even if recent studies have been already cited.

Professional article structure, figures, tables. Raw data shared.

The article is separated in standard sections. No table is presented. Thanks to the authors to share raw data. Figure 3 could be enhanced since it is difficult to visualize it. A higher resolution is needed, the plots do not have the same size and the text is small.

Self-contained with relevant results to hypotheses.

The presented results answer to the hypotheses and questions asked.

Experimental design

Original primary research within Aims and Scope of the journal.

The question is pretty well described. It appears relevant even if according to me, the link with fitness could be more developed. The topic fits with the aims and scope of the journal.

Research question well defined, relevant & meaningful. It is stated how research fills an identified knowledge gap.

The research question is correctly defined but the novelty of the study could be improved. Does this study bring new insights of the understanding of climate change? I think it is the case but it is not really developed in the introduction and discussion parts. Why the species is peculiarly relevant to answer to this question ? The knowledge gap concerning the question should be more deeply developed.

Rigorous investigation performed to a high technical & ethical standard.

The experiment has been conducted in conformity with ethical standards in the field. The experimental design is impressive, even if the number of fish by group is low but the experimental design is strong.

Methods described with sufficient detail & information to replicate.

Methods are well described but there is a lack of information on some parts:
1) Concerning the study species, providing information about thermal limits of the species could be interesting. Why this species is interested to answer this question?
2) In the warming exposure part, I was wondering why individuals were kept during 2 weeks in the habituation phase
3) I am not familiar with behavioral studies but how time delays for behavioral experiments have been choosen ? It lacks some literature to justify these choices (L166-L188).
4) Concerning statistical analyses, there is a lack of information about models. I would be interested in the details of models, which effects have been used as fixed, as random ones? Are there covariates?

Validity of the findings

Impact and novelty not assessed. Meaningful replication encouraged where rationale & benefit to literature is clearly stated.

Results show interesting results in regard of the question of climate change on behavior in marine fish. In relation with what I said previously, novelty of the study could be highlighted in the introduction by describing the gap in knowledge about this question and how the study brings new findings. This could also be developed in the discussion part more deeply.


All underlying data have been provided; they are robust, statistically sound, & controlled.

Thanks to the authors to have provided data. I mention that the number of individuals per group is pretty low (10 individuals). But I am aware of the fact that behavioral experiments are pretty heavy in logistical terms and I understand it is difficult to process a lot of individuals. Perhaps, it could be interesting to justify this number of individuals per group.

Conclusions are well stated, linked to original research question & limited to supporting results.

The discussion is interesting and well written. To me, there is a lack of development about the link with fitness and consequences for fish survival. It is well described in the introduction part but the link is not done in the discussion part. Even if it is correlative, I think it could enhance the paper to integrate this part. It could be developed in the last part of the discussion, in the perspective part, in the place of the one developed here.

Additional comments

Figure 1. There is a lack of space between “and” and “Amphiprion”.
Figure 2. For me, the Figure 2 is not very clear. First, I don’t understand the expectations about the “hall 2”,” hall 3” and “hall4”. Moreover, the figure does not show the effect of temperature on behavioral observations since the three levels of temperature are separated. I would suggest to try to present the temperature on axis x, so as to make the link with the results part (link with the model testing the effect of temperature)
L191: “No other strategy” instead of “no other strategies” ?
L200-201: “No animal was excluded” instead of “no animals were” ?
L294-296: Rephrase the sentence please
L302: A coma after “Here”
L316: “the impact of warming ON cognition” instead of “to cognition”

Reviewer 3 ·

Basic reporting

I found the article well written and well referenced. The introduction reads very nicely and sets a good background for the study.

Experimental design

The research question is well defined and relevant, and the experimental design is appropriate to investigate it. The methods are well written and very detailed.

However, I have one specific concern as to how learning is assessed. The authors use latency measures, particularly the latency to enter the target hall and the time spent in the target hall, as the measures for learning ability. I am wondering why they have not scored which hall the fish chooses to enter first. Scoring choice is typically the best estimate for learning. Latency to choose & time spent in target area are known to not be very accurate estimates of learning. First, in this specific case I think both latency and time spent in specific areas can be confounded if the temperature treatment affects traits such as exploration, as mentioned in the discussion. I acknowledge that the authors report no effects on locomotor ability, which could also be a confounding effect on latency measures, but I think it still doesn't eliminate effects on personality or other behavioural traits that are not learning per se. Secondly, there is a vast literature on speed-accuracy trade offs where often animals that show high choice accuracy display slow latency to respond, for example, indicating that latency is not a good predictor of learning in many cases. Since the authors video recorded the trials, I would highly recommend to score which hall the fish enters first to analyse choice accuracy in addition to latency.

Validity of the findings

Data have been provided. Conclusions are well stated and discussion is well written. However, as per my comment above, I think the conclusions would be stronger if accuracy was added as a measure of learning, in addition to the latency measurements.

Additional comments

I would suggest a few edits to improve clarity.

- L 44, please define 'stenothermic'

- Overall, I think there is some confusion with the terminology 'stimulus fish'. This fish is actually the reward in this task, a social reward. Your stimulus is the blue cue card, that the fish needs to learn to associate with the reward. Please replace 'stimulus fish' with 'social reward' (or another designation you prefer)

- I would suggest rephrasing most of the results section to highlight the behaviour of the fish and what you found, instead of the statistical tests. As it is, I found it very hard to understand how the fish in the different treatments varied. For example, L.215 "Comparisons made by the mixed model indicated statistical significance caused by water temperature (GLMM, X² = 6.35, df = 2, p =0.042)" -> we found that the fish in each water temperature group differed in the percentage of time spent in each hall of the tank. In particular, the control group spent more time in the target hall on both the learning and memory tests than the high warming group (Tukey posthoc tests, p=0.035).

- I would also suggest changing how you visually represent your results. I think bar plots are a bad choice to depict data in general, but especially with low sample sizes.
In fig 2 and 4, I suggest using boxplots which give more information.
But my main issue was with Figure 3, which I personally found very hard to read. I have a few suggestions that I think could improve its readability. I recommend to separate figure 3 into several panels. First, the training days are a sequence of temporal data that you analyse with day as a fixed effect. So why not show regression lines of how latency changes over time for the training days? You can have one plot with the three temperature lines. And since the learning test and memory test are slightly different protocols, and your aim is to compare the tree treatments in these measures, why not have the learning test and memory test as independent plots and show the three treatments side by side? It is very hard to visually compare the control with the high temperature group and see which one was faster/slower, when they are vertically aligned.

---

## Round 0.2 · Minor Revisions

We have received a review from a previous reviewer on your manuscript. As detailed below, the reviewer thinks that the interpretation of the main findings may be affected by a confounding issue that is still not adequately addressed in this revised version.

More specifically, the reviewer pointed out that reduced learning opportunity, of warm-acclimated fish, may explain the results, rather than a direct effect of warming on cognitive ability. I thus strongly encourage the authors to rework their manuscript to further recognize this possible limitation.

Comments regarding data presentation should also be taken into account.

Reviewer 1 ·

Basic reporting

OK

Experimental design

See Additional Comments

Validity of the findings

See Additional Comments

Additional comments

I am pleased to see that the authors have responded to the concerns originally raised by the referees. I am largely satisfied with these responses, but I do have some additional concerns.

My main concern continues to stem from the fact that not all fish received the same amount of training during the 5 associative learning days. I appreciate that these data are now briefly mentioned, but in my opinion this issue should be brought up in the discussion section. I think this is an important issue that deserves exploring in full paragraph. It is not clear why warm-acclimated fish took longer to first discover the target hall during the learning period (the obvious biological explanations about differences in swimming speed/activity are ruled out in Figure 5), but this could be a major driver of the differences presented in Figs 3B and 4. This is important, because the major conclusions of the study stem from those data. In other words, temperature might not impair learning/memory directly, but rather the results may stem from a shorter learning period in warm-acclimated fish. This should be acknowledged.

Regarding the latency data presented in figure 3 – I am pleased to see the data in 3b presented as boxplots, but I think that presenting this data as a function of experimental day (i.e. as it was in Fig3 of the original draft) would be preferable. In my opinion, showing the change in latency over time illustrates the best evidence for learning (in the control group), and would recommend it be retained. This would also help clarify which latency data is plotted for “learning” – it’s unclear which of the learning trials is shown here. Overall, it may be helpful to show all the data as a function of time (e.g. also the zone data show in Fig 2, and the first choice data shown in fig 4). Finally, despite several attempts, I was unable to understand the importance of Figure 3A. I suggest clarifying the text in the methods/results to aid interpretation of this panel, or removing it altogether.

Regarding Figure 4 – I think these “first choice” data are valuable to include given the objective of the experiment. However, boxplots are a good way to show these data. Boxplots are ideally suited for continuous data, not discrete data as presented here. In addition, I would recommend combining the non-target data (i.e. Halls 1-3) similarly to Figure 2. I don’t see any useful reason to separate out choices between these three halls, and the comparison would be much easier to visualize if it was simplified to target/non-target. Finally, as mentioned above, it’s unclear which of the learning trials is shown in panel B.


Other comments
Line 69 Brownscombe spelling
Line 229 – not clear to me why “the number of training trials needed 230 to find the reward” is a good indicator of learning
Line 275 I suggest removing your interpretation “…confused the individuals…” from the Results text.

---

## Round 0.3 · accepted · Accept

I am generally pleased by the final revisions made to the manuscript. I would only suggest the following minor revisions:

L367 Salvanes & Braithwait, 2006 should be Salvanes & Braithwaite, 2006

L372: remove ‘what help then to cope more effectively’